

# Tracking multiple fish

Filip Děchtěrenko[1], Daniela Jakubková[1], Jiří Lukavský[1] and
Christina J. Howard[2]

[1] Institute of Psychology, Czech Academy of Sciences, Prague, Czech Republic
[2] Nottingham Trent University, Nottingham, United Kingdom

## ABSTRACT

Although the Multiple Object Tracking (MOT) task is a widely used experimental method for studying divided attention, tracking objects in the real world usually looks different. For example, in the real world, objects are usually clearly distinguishable from each other and also possess different movement patterns. One such case is tracking groups of creatures, such as tracking fish in an aquarium. We used movies of fish in an aquarium and measured general tracking performance in this task (Experiment 1). In Experiment 2, we compared tracking accuracy within-subjects in fish tracking, tracking typical MOT stimuli, and in a third condition using standard MOT uniform objects which possessed movement patterns similar to the real fish. This third condition was added to further examine the impact of different motion characteristics on tracking performance. Results within a Bayesian framework showed that tracking real fish shares similarities with tracking simple objects in a typical laboratory MOT task. Furthermore, we observed a close relationship between performance in both laboratory MOT tasks (typical and fish-like) and real fish tracking, suggesting that the commonly used laboratory MOT task possesses a good level of ecological validity.

## INTRODUCTION

Past research has extensively shown that people are able to successfully track several moving circles within a specified arena. This task is known as Multiple Object Tracking (MOT; *Pylyshyn & Storm, 1988*) and it has been used for various research questions ranging from the theoretical mechanism behind tracking (*Pylyshyn & Storm, 1988*; *Yantis, 1992*; *Cavanagh & Alvarez, 2005*), position extrapolation *vs* lag (*Howard, Masom & Holcombe, 2011*; *Luu & Howe, 2015*), and eye movements (*Zelinsky & Neider, 2008*; *Lukavsky, 2013*; *Fehd & Seiffert, 2008*). However, in the real world, we almost never encounter conditions that are highly similar to this laboratory task.

In the real world, we rarely encounter cases where objects cannot be recognized by their visual features. All items in the world have an identity and when we track multiple objects, this identity can help us to recover lost objects, and the tracking periods are usually only brief. Conversely, in typical laboratory-based MOT tasks, objects are usually indistinguishable and participants need to rely on attentional tracking only (*Pylyshyn & Storm, 1988*). Several studies show that participants take advantage of surface features of objects (*Makovski & Jiang, 2009*; *Papenmeier et al., 2014*; *Zhao et al., 2020*;

Corresponding author
Filip Děchtěrenko,
dechterenko@praha.psu.cas.cz

*Horowitz et al., 2007*). In particular, when targets and distractors can be distinguished by some visual feature (even when this feature is not available in the query phase), participants were more accurate than when targets and distractors possessed the same visual features (*Horowitz et al., 2007*; *Papenmeier et al., 2014*).

An additional difference between real-world tracking and laboratory MOT tasks is the variable difficulty of object detection in real-world contexts. In the real world, objects often move against non-uniform backgrounds and thus can be difficult to visually detect. Although moving objects are easier to detect than static ones (see literature regarding visual search and motion, *e.g.*, *Royden, Wolfe & Klempen, 2001*; *Ivry & Cohen, 1992*), it is rational to assume that difficulty in detecting targets due to contrast masking would impair tracking performance. Of course, objects that are camouflaged to some extent against a background are often more difficult to detect (*Sharman, Moncrieff & Lovell, 2018*; *Troscianko et al., 2013*). However, in the camouflage literature, although moving camouflaged objects are easy to detect, they are more difficult to identify (*Hall et al., 2013*). To our knowledge, the impact of lower detectability on tracking performance in the laboratory and in the real world is not well understood. The third important difference is the variability in movement trajectories. In MOT tasks, objects usually follow simple movement patterns such as ballistic or Brownian motion. Differences in movement patterns are an important factor influencing tracking performance because sudden changes in movement can attract attention (*Howard & Holcombe, 2010*), and less predictable trajectories can lead to reduced tracking performance (*Ericson & Beck, 2013*; *Howe & Holcombe, 2012*). Moreover, when the object speed is not constant, tracking performance is also reduced (*Meyerhoff et al., 2016*). All these factors are common in movements observable in the world. Objects can suddenly stop and change direction rapidly, and perfectly constant speed and direction of motion are rarely observed.

As mentioned in a recent review by *Meyerhoff, Papenmeier & Huff (2017)*, a crucial future direction in the MOT literature is to strengthen its ecological validity. Some progress has been made in recent years. For example, multiple object tracking has been used in studies on dividing attention during driving (*Ericson et al., 2017*; *Lochner & Trick, 2014*) and several studies explored the relationship between sport expertise and attentional tracking (*Memmert, Simons & Grimme, 2009*; *Qiu et al., 2018*; *Howard, Uttley & Andrews, 2018*).

One way to increase ecological validity is to use stimuli that are more similar to those found in real life. *Zelinsky & Neider (2008)* did just this and used 3D rendered sharks swimming in the sea as stimuli. However, there were several differences between their task and the equivalent real-world scenario: In their study, sharks could be easily detected against the background and possessed a fairly uniform texture. Furthermore, the sharks moved randomly at constant speed. In the real world, nautical creatures usually follow more variable movement patterns (*Huth & Wissel, 1992*; *Huth & Wissel, 1994*). As such differences in motion characteristics have potential impacts on ecological validity, further research is required to investigate this issue.

In this project, we evaluated the ecological validity of MOT tasks by comparing performance with traditional MOT stimuli and with actual stimuli that we can encounter
in the real world. For that purpose, we filmed fish in Sea World for several hours and used the recorded videos as stimuli. We explored performance in the MOT task using fish as objects (Experiment 1). In Experiment 2, we estimated individual variability in tracking when performing this fish tracking MOT task, in standard MOT and a third condition in which standard circle stimuli were used whose motion mimicked that of the fish. This third condition was introduced as one that is intermediate in its characteristics between standard tracking and the novel fish tracking task.

## EXPERIMENT 1

The purpose of Experiment 1 was to (1) observe, to what extent participants are able to track moving fish in an aquarium, and (2) estimate the difficulty of the task.

All experiments were approved by the Ethics Committee of the Institute of Psychology, Czech Academy of Sciences (approval number PSU-965/Brno/2021).

## MATERIALS AND METHODS

### Participants

Ten students (mean age: 22.2 years, SD = 1.48 years, three men) participated in an hour-long experiment in exchange for course credit. Participants signed the informed consent form prior to the experiment. All subjects reported normal or corrected-to-normal vision. Up to five participants were measured at the same time.

### Stimuli and apparatus

Stimuli were short videos of fish moving in an aquarium. We filmed 6 h of movement, from which we created 23 short videos (each 10 s long). The number of fish in each video varied (15–17 fish per video). We noticed that the fish alternated between fast movement around the scene with intervals in which they remained at one location for a while with a small positional jitter. There were three rocks at the bottom of the aquarium, a water purifier in the upper part of the aquarium, and one artificial plant. The aquarium had a rocky background texture. All of these factors decreased the detectability of fish in several areas of the scene. The speed and density of the objects (fish) on the screen varied between trials. The images and videos were presented on a 22″ LCD screen monitor with 1,920 × 1,200 pixel resolution (resolution of the presented video was 960 × 600 px).

### Procedure and design

The experiment was prepared in PsychoPy (*Peirce et al., 2019*). In all trials, the participants' task was to track several fish among others. Each trial followed a similar structure. First, a white fixation cross was displayed at the center of the screen for 75 ms, followed by a static cue phase showing the first frame of the video with highlighted fish as targets. The targets (either one, three, or five fish) were highlighted by superimposing a yellow ellipse for 3 s. After the cue phase, the cues disappeared, and the fish began to move. After 10 s, all the fish stopped and one of the fish was highlighted (with the probability of this object being a target set at a constant 50%). The participants then decided whether the highlighted fish was among the original targets (left arrow key) or not (right arrow

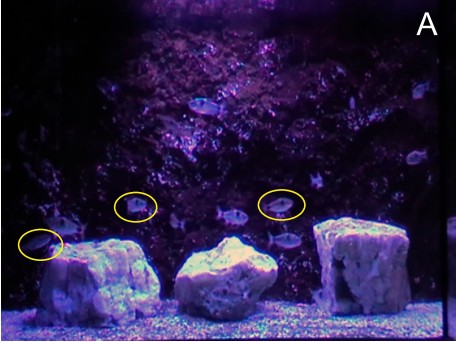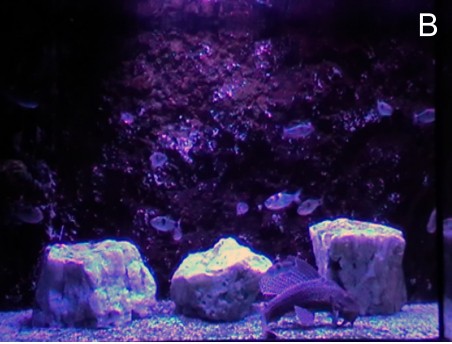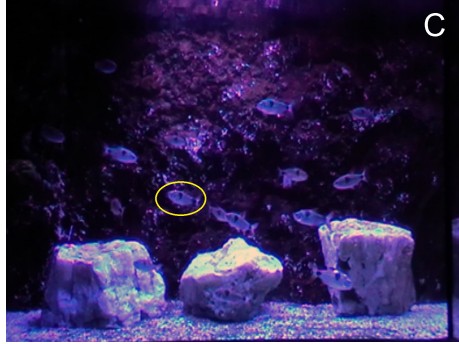

**Figure 1 Schematic representation of the trial.** The cue phase with three targets (A) was followed by 10 s movie stimuli with moving fish (B), and finally one fish (either target or distractor) was queried (C).

key). Participants had unlimited time to respond, and text cues with key arrows were displayed below the video. Sample frames of each phase are visualized in Fig. 1.

Each participant undertook and completed a total of 138 experimental trials. From each video, we created six video stimuli for use in six different trials: two for each set size (one, three, or five targets), and for each set size, one trial where a distractor was queried and one in which a target was queried, thus resulting in six total presentations of each video. The order of the trials was randomized for each participant.

## Data analysis

All materials and scripts are available at the Open Science Framework (https://osf.io/tfrzj/). Data was analyzed with R (*R Core Team, 2020*) and brms package (*Bürkner, 2017*). We modelled how performance was affected by tracking load (number of targets as a linear predictor). We describe the tracking performance in terms of the signal detection theory measures of sensitivity (d') and bias. Using signal detection theory measures offers an advantage over simple averages of correct responses, as it not only models the sensitivity to correctly detect the previously seen target, but also the response strategy of individual participants. In particular, estimates of bias tell us whether participants are more likely to mark a queried object as target (negative values) or as distractor (positive values).

Both measures were calculated *via* the multilevel Bayesian logistic regression model with probit link function (*DeCarlo, 1998*; *DeCarlo, 2010*). Use of logistic regression for estimating signal detection theory measures offers several advantages over using typical approach that estimates the sensitivity and bias based on the averaged data. Estimated parameters of fitted logistic regression correspond to sensitivity and bias as shown by *DeCarlo (1998)* and therefore it is simple to extend this approach to more complex designs with hierarchical structure of data or to add more parameters into the model. In the model, group-level parameters included individual slopes and intercepts for each participant. We used weakly-informative Gaussian priors ($N(0,1)$) for both intercept and slope parameters. In the text, we report posterior means and 95% credible intervals.

In the next step, we evaluated the influence of three other factors (fish count, number of sudden motions in targets, number of sudden motions in distractors). We included each of the predictors separately in the model described above and explored their effect on

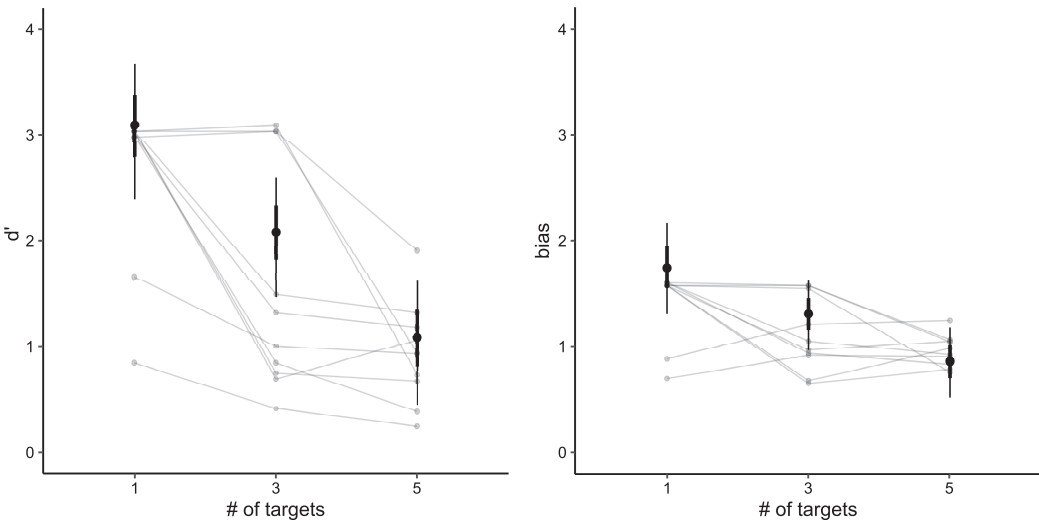

**Figure 2 Posterior predictive distributions for both sensitivity (left) and bias (right) based on the number of targets.** Vertical lines denote credible intervals (thin line: 95%, thick line 66%). Connected gray points represent average values per individual participant.

sensitivity and bias. We report Bayes factors to indicate the evidence regarding whether the respective parameters differ substantially from zero.

## RESULTS

The results showed that tracking accuracy decreased with the number of targets from 95% (SD = 7%) for one target, to 81% (14%) for three fish to 73% (11%) for five fish targets.

Our model showed that participants were most sensitive when tracking a single target (d′ = 3.08, 95% CI [2.47–3.73]). Their sensitivity decreased with additional targets (three targets: d′ = 2.08, 95% CI [1.51–2.63]; five targets: d′ = 1.07, 95% CI [0.45–1.63]). In general, people were more likely to make false alarm errors (positive bias values). The highest bias was in the case of one target (bias = 1.75, 95% CI [1.33–2.19]), and decreased for three targets (bias = 1.30, 95% CI [0.98–1.63]) and five targets (bias = 0.86, 95% CI [0.52–1.18]).

Analogously, we can say that adding a single fish decreases sensitivity on average by −0.50 (95% CI [−0.62 to −0.39]). For bias, the addition of a single fish decreased the bias by −0.22 (95% CI [−0.33 to −0.13]). Figure 2 shows the posterior distribution for sensitivity and bias (including sensitivity and bias for individual participants).

Additional information about the scene and objects' movements had only a very limited effect on tracking performance. For the number of fish in the scene, our data indicate support for no influence and the estimated effects on sensitivity and bias were close to zero (d′: $BF_{10}$ = 0.399; bias: $BF_{10}$ = 0.134). There is strong evidence that sudden movements of targets affect sensitivity ($BF_{10}$ = 18.31). As expected, the effect is negative: the odds ratio decreased by 0.87 (95% CI [0.80–0.95]) for each sudden motion change. Motion changes in targets had no effect on bias ($BF_{10}$ = 0.127). Sudden movements of distractors had no effect on sensitivity ($BF_{10}$ = 0.026) or bias ($BF_{10}$ = 0.023).

## INTERIM DISCUSSION

In this experiment, we explored the tracking performance in a highly ecologically valid setting. The results suggest that the performance of this task shares commonalities with the performance of the classic MOT task in terms of load effects. Signal detection measures showed an approximately linear decrease in sensitivity with load, with performance remaining above floor levels. The changes in bias show that people take the proportion of targets to all objects into account in their responses. When asked to track a single target they tend to err in favor of 'miss' responses (marking a target as a distractor) perhaps indicative of participants' correct belief that the probability of any single object being a target is lowest in trials with lower numbers of targets. In fact, the probability of any queried object being a target remained constant at 50% but participants were not told this. With more tracked targets (five of approx. 15) the bias decreased although it was still present. There are several aspects that make this tracking task different from traditional MOT. First, the fish moved within an aquarium that contained various objects, which allowed the fish to blend in or even to be temporarily occluded, making them more difficult to detect. This corresponds to findings that tracking objects with temporary occlusions leads to reduced performance (*Scholl & Pylyshyn, 1999*; *Horowitz et al., 2006*; *Lukavsky, Oksama & Dechterenko, 2022*) though people appear to be able to track objects under a variety of circumstances in which targets disappear and reappear (*Keane & Pylyshyn, 2006*). Second, we used more objects (16 on average) than is common in MOT. Having more objects increases the chance of crowding, and again makes the task more difficult. Finally, in common MOT tasks, objects usually move continuously, without waiting at one spot, whereas in this experiment, fish usually waited at one spot and then made a sudden movement. Evaluation of our models suggests that these sudden movements make the task more difficult.

## EXPERIMENT 2

Experiment 1 revealed that performance when tracking fish in an aquarium has similarities with traditional MOT. In Experiment 2, we focused on individual differences and tested the extent to which performance in a traditional MOT task can be used to predict the performance in a naturalistic task (tracking fish). To further examine the potential role of motion differences between fish motion and object motion in traditional MOT, we introduced an additional condition where objects were visually similar to those used in classic MOT but their motion mimicked that of real fish. We refer to this condition as fakefish-MOT and compared it with a traditional-MOT and fish-MOT conditions. To make the distinction between naturalistic task and laboratory versions clearer, we refer to both traditional-MOT and fakefish-MOT as circle-tracking MOT. Each participant was measured in all three conditions to further assess the ecological validity of the classic MOT task.

## MATERIALS AND METHODS

### Participants

Fifty-three students participated in the study in exchange for course credit. All students had normal or corrected-to-normal vision, and none of them participated in Experiment 1.

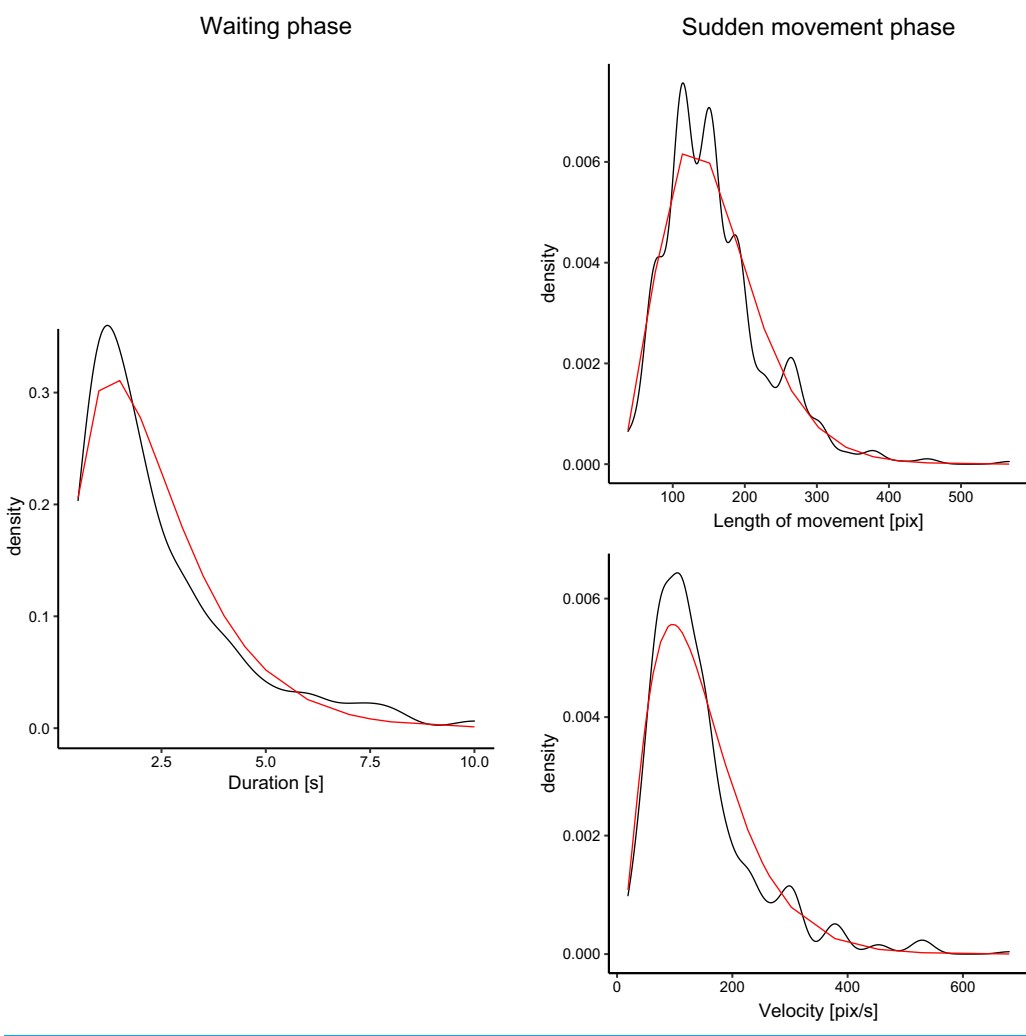

**Figure 3  Distribution of the duration of waiting times (for waiting phase), and velocity and distances (for sudden movement phase).** The black lines denote density plots of the observed fish data, and the red line denotes the fitted gamma distribution.

Participants gave their informed consent prior to the experiment. One participant was presented with the same condition twice by error and was removed from the analysis, resulting in the final sample consisting of 52 participants (mean age: 21.6 years, SD = 3.51 years, all females). One participant did not fill in their age.

## Fish movement analysis

We analyzed fish movement patterns in all videos. For each fish, we decomposed the trajectory into waiting phases (fish remained approximately at the same position-within an area of 60 px radius) and sudden movement phases, in which fish moved into different locations in the aquarium. We measured the durations of waiting phases, and distances and directions for sudden movement phases. The fish spent approximately 19% of all time waiting and 81% moving. For all measurements of each parameter (*e.g.*, waiting times), we fitted the data with the gamma function using the maximum likelihood estimate. The fit

for waiting times, velocity, and distances is visualized in Fig. 3. For all three parameters, the gamma functions fitted the distribution well with the exception of the peak, which was underestimated in comparison to the ground truth. However, our goal was not to create a precise model of fish movement but rather to create moving patterns that displayed some level of similarity with real fish movement.

## Stimuli and apparatus

The apparatus was the same as for Experiment 1. In this experiment, there were two tracking loads: two or four targets per trial. For fish-MOT, we used the same videos as in Experiment 1. For the two circle-tracking MOT conditions (fakefish-MOT and traditional-MOT), we used grey circles subtending approximately 1°. In the cue phase, all target circles changed color to green, while the color of the remaining objects remained midgray. In the query phase, the queried object changed color to yellow. Objects moved in a circular arena with a diameter of 800 px. This size was selected to approximately match the area of fish motion in the aquarium videos when displayed on the screen ($960 \times 600$ px). In the traditional-MOT, objects moved with constant speed 200 px/s (approximately 6°/s). In both circle conditions, objects bounced off borders of the circular arena, and in traditional-MOT, they additionally bounced off each other. We did not introduce such bounces in the fakefish-MOT since the real fish would not do this, and we wanted to increase similarity between fish-MOT and fakefish MOT motion characteristics. For circle-tracking MOT, we pregenerated objects' trajectories using the motrack package (*Lukavsky, 2021*).

## Procedure

Each participant was presented with three different conditions: tracking real fish (fish-MOT), tracking fake fish (fakefish-MOT), and tracking objects in typical MOT task (traditional-MOT). The order of conditions was counterbalanced between participants (six different orderings of blocks). In both circle-tracking MOT tasks (fakefish-MOT and traditional MOT), participants completed four training trials, which were discarded prior to analysis. In all trials, the moving phase lasted 10 s.

The real fish trials were similar to Experiment 1 except that the tracking load was set at either two or four fish per trial. Participants in Experiment 2 completed 92 trials in each condition (two difficulty levels with 2/4 targets uniformly balanced) experimental trials. Each participant was presented with the same set of trajectories. For real fish, we used the same video in two and four target conditions, while in circle-tracking MOT conditions, we used unique trajectories for each condition. The order of the trials was different for each participant. The whole experiment lasted approximately 75 min.

## Data analysis

As in Experiment 1, we used Bayesian modelling to answer two questions. First, what are the overall performance differences between the conditions? Second, can we predict sensitivity in one condition based on performance in the other conditions at the individual level?

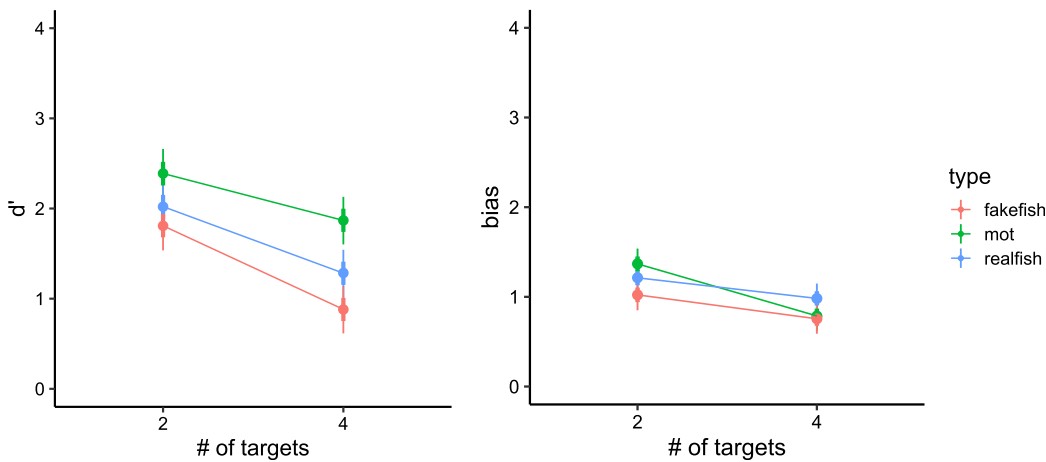

**Figure 4** **Model estimates for signal detection theory measures (left panel: d′, right panel: bias) in all three conditions.** Thick vertical lines denote 66% high density credible interval, thin vertical lines denote 95% high density credible interval.

For the first question, we modelled sensitivity and bias *via* multilevel logistic regression models with probit link function based on number of targets and condition (traditional-MOT, fish-MOT, and fakefish-MOT, deviation coded) similarly to Experiment 1. Essentially, we extended the model from Experiment 1 by adding the condition parameter and interaction with the number of targets. We also report pairwise comparison of sensitivity and bias between three conditions. To compare the parameters, we calculated the difference in the posterior estimates from trials with two and four targets for given conditions and report credible interval for this difference.

Second, we modelled the results at the individual level and attempted to predict the sensitivity in fish-MOT either by predicting it from the sensitivity in traditional-MOT or from the sensitivity in fakefish-MOT. Here, we averaged sensitivity/bias per participant and used multilevel Gaussian models. Again, we added an additional factor denoting number of targets (including the interaction) and we quantified the added value of both main factor and the interaction using Bayes factors. To assess the fit of both models, we used the Bayesian version of the traditional $R^2$ coefficient (*Gelman et al., 2019*).

In all models, we used weakly-informative Gaussian priors ($N(0,1)$) for both intercept and slope parameters.

As an attention check, we removed participants whose average accuracy in traditional-MOT with two targets was lower than 50% (the two-target condition is usually considered an easy task). This cutoff criterion led to the exclusion of two participants. To ensure that this removal did not change the results, we also ran the analysis on the original sample and obtained similar results.

## RESULTS

In general, by pooling between trials with two and four targets, the participants achieved the highest accuracy in traditional-MOT (83%, SD = 16%), followed by realfish-MOT (77%, SD = 16%) and fakefish-MOT (73%, SD = 14%). As visualized in Fig. 4, in all three

conditions, sensitivity and bias decreased in trials with four targets compared to trials with only two targets.

In particular, our models showed that in traditional-MOT the sensitivity in trials with two targets was greater (d′ = 2.39, 95% CI [2.13–2.67]) than in trials with four targets (d′ = 1.87, 95% CI [1.60–2.13]). The same pattern was observed in realfish-MOT (two targets: d′ = 2.02, 95% CI [1.76–2.30]; four targets: d′ = 1.28, 95% CI [1.02–1.55]) and fakefish-MOT (two targets: d′ = 1.81, 95% CI [1.54–2.07]; four targets: d′= 0.88, 95% CI [0.62–1.14]). To compare sensitivity between conditions, we pooled the data for two and four targets. Traditional-MOT showed higher sensitivity than both fakefish-MOT (Δd′ = 0.78, 95% CI [0.43–1.13]) and realfish-MOT (Δd′ = 0.48, 95% CI [0.21–0.72]). The sensitivity in realfish-MOT was higher than in fakefish-MOT (Δd′ = 0.31, 95% CI [0.07–0.53]).

In the case of bias, our models showed the highest bias (more false alarm errors) in traditional-MOT (two targets: bias = 1.37, 95% CI [1.19–1.53]; four targets: bias = 0.79, 95% CI [0.63–0.95]), followed by realfish-MOT (two targets: bias = 1.21, 95% CI [1.05–1.40]; four targets: bias = 0.98, 95% CI [0.82–1.15]) and fakefish-MOT (two targets: bias = 1.02, 95% CI [0.84–1.18]; four targets: bias = 0.76, 95% CI [0.61–0.93]). Again, pairwise comparison of posteriors distributions in case of bias revealed that bias was smaller in fakefish-MOT than in both traditional-MOT (Δ bias = 0.19, 95% CI [−0.07 to 0.45]) and realfish-MOT (Δ bias = 0.21, 95% CI [0.09–0.33]), while bias in traditional-MOT and realfish-MOT was similar (Δ bias = −0.02, 95% CI [−0.30 to 0.02]). These results suggest that the increase in false alarms at lower loads is similar in all three conditions. The pairwise comparison of pooled data showed that bias was lowest in fakefish-MOT. In another words, in more difficult tasks (with more targets or in the fakefish condition), participants were more likely respond that the target was a distractor.

When predicting sensitivity in fish-MOT using sensitivity in traditional-MOT (Fig. 5), we observed a large estimate that excluded zero (posterior mean = 0.83, 95% CI [0.46–1.19], $BF_{10}$ = 551,381). The estimate of the effect of the number of targets included zero (posterior mean = −0.00, 95% CI [−0.21 to 0.20], $BF_{10}$ = 0.095), as did the interaction between the number of targets and the sensitivity in traditional-MOT when predicting the sensitivity of fish-MOT (posterior mean = −0.14, 95% CI [−0.28 to 0.00], $BF_{10}$ = 0.57). For bias, all estimates were small with credible intervals that included zero (bias in MOT: posterior mean = 0.13, 95% CI [−0.43 to 0.68], $BF_{10}$ = 0.317; number of targets: posterior mean = −0.05, 95% CI [−0.25 to 0.14], $BF_{10}$ = 0.110; interaction: posterior mean = 0.04, 95% CI [−0.17 to 0.26], $BF_{10}$ = 0.116). These results suggest that the response bias of participants in traditional-MOT cannot be used to predict the response bias in fish-MOT.

When predicting the sensitivity in fish-MOT from the sensitivity in fakefish-MOT (Fig. 6), we observed estimates similar to those using traditional-MOT (posterior mean = 0.88, 95% CI [0.40–1.35], $BF_{10}$ = 67.1). Again, the estimate for the number of targets was close to zero (posterior mean = 0.01, 95% CI [−0.17 to 0.20], $BF_{10}$ = 0.089). In this case, the interaction also showed a lack of effect (posterior mean = −0.04,

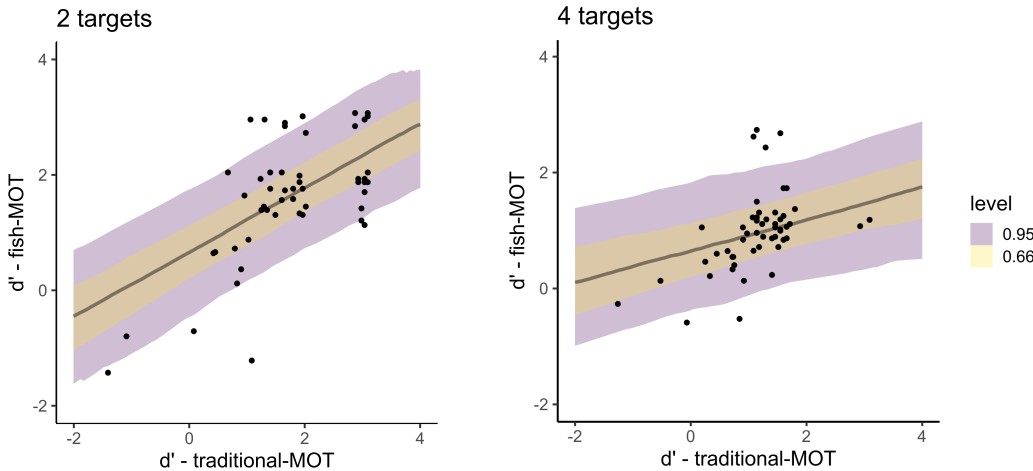

**Figure 5 Posterior distributions that capture the prediction of the sensitivity in fish-MOT based on the sensitivity in traditional-MOT in the case of two targets (left) and four targets (right).** The color of the band specifies the probability for a given credible interval (95% or 66%). Each dot represents one participant.

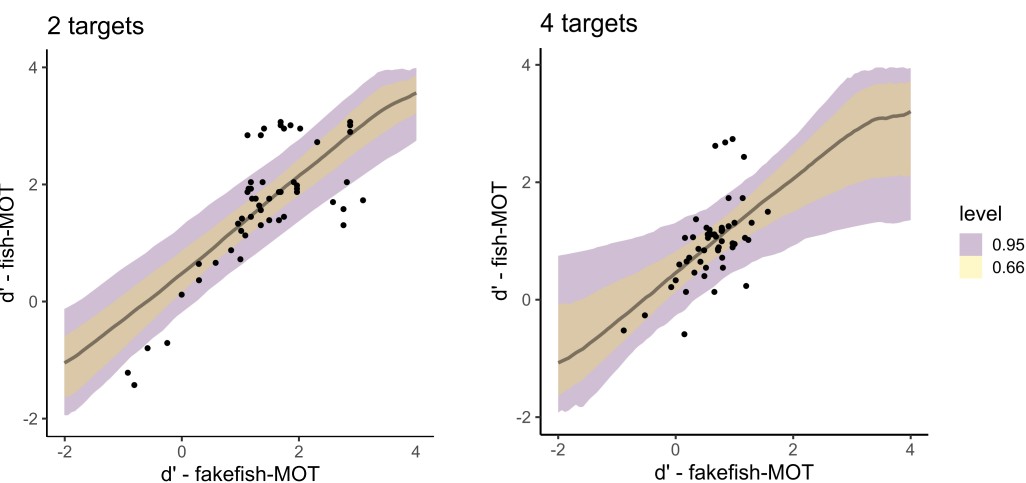

**Figure 6 Posterior distributions that capture the prediction of the sensitivity in fish-MOT based on the sensitivity in fakefish-MOT in the case of two targets (left) and four targets (right).** The color of the band specifies the probability for a given credible interval (95% or 66%). Each dot represents one participant.

95% CI [−0.24 to 0.17], $BF_{10} = 0.106$). For bias, we again found that bias in fish-MOT cannot be predicted by bias in fakefish-MOT ($BF_{10}s \leq 0.31$).

The fit of both models predicting sensitivity was expressed using Bayesian $R^2$, which is the Bayesian equivalent of traditional $R^2$ estimates of model fit. Both models showed similar fit (predicting fish tracking performance from traditional MOT: $R^2 = 0.70$, 95% CI [0.53–0.80], predicting from fakefish-MOT: $R^2 = 0.69$, 95% CI [0.55–0.80]). In other words, both circle tracking MOTs could predict the performance of real-world MOT task.

## GENERAL DISCUSSION

The Multiple Object Tracking task is one of the most commonly used paradigms for studying divided attention. In this study, we explored the ecological validity of the task by using more naturalistic stimuli–real fish. Using fish has clear advantages, as they are difficult to identify from each other and thus the task shows resemblance to stimuli used in most Multiple Object Tracking studies. We found that using fish as stimuli leads to a task with intermediate difficulty–the majority of participants did not reach the upper or lower bound on tracking performance. Furthermore, we observed decreases in tracking accuracy when the number of tracked fish increased. This effect is robustly observed in multiple MOT studies (*Pylyshyn & Storm, 1988*; *Meyerhoff, Papenmeier & Huff, 2017*).

The main finding of the study is the close relationship between the performance in circle-tracking MOT and performance in tracking real fish. This suggests that MOT tasks commonly used in laboratory research do show resemblance to tracking in real world contexts. This correspondence appears to be valid for situations where human observers cannot take advantage of surface features or identities of tracked objects. This is true for tracking fish (or other groups of hard-to-distinguish animals), but further research must be done in other common real-world conditions like tracking playing children, where perhaps greater use of identity tracking may be used, a task commonly mentioned as one example of tracking in the wild.

The findings presented here show the importance of using more ecologically valid movement trajectories. As observed by others (*e.g.*, *Ericson & Beck, 2013*), movement patterns are an important factor influencing tracking accuracy. Using trajectories mimicking movement patterns of real fish leads to even higher intra-individual correspondence between tracking accuracies for real and artificial stimuli. When we decomposed the movement pattern of the fish, we found intermixed intervals, when the fish stayed in one place or possessed sudden fast movement, and such motion patterns are not often adopted for stimuli in laboratory settings. These sudden fast movements were one predictor that influenced the probability of losing the target. Past research has shown that sudden motion onset captures attention (*Christ, 2003*), and thus attention could be withdrawn from other targets, resulting in decreased accuracy overall. Additionally, these sudden accelerations were accompanied by quick fast directional changes, which may also capture attention (*Howard & Holcombe, 2010*). Alternatively, one possibility is that this may be due to the fact that the sudden fast movements of targets also increased their average speed, distance traveled, and thus also the number of close encounters with distractors, which has been suggested to be a main factor in tracking accuracy (*Franconeri, Jonathan & Scimeca, 2010*).

Using artificial fish trajectories showed similar intra-individual correspondence between tracking accuracies using real and artificial stimuli. In both cases, there was still some variance unaccounted for. One difference between real fish and laboratory conditions was the presence of the rich visual background in the case of real fish stimuli. The fish could occasionally blend with the sandy water bed and the visual system needed to detect the object as a prerequisite to the actual tracking. The effect of reduced detectability should be explored in further studies.

## CONCLUSIONS

We found that performance in the standard MOT task can predict accuracy of tracking in more real-world settings, arguing the case that MOT shows good signs of ecological validity. Although using motion trajectories in MOT that are derived from a real-world context of fish movement led to similar predictions of performance for tracking real fish, future studies should further explore the importance of motion characteristics in predicting real world tracking.

### Funding

This work was supported by the Czech Science Foundation under Grant GA 19-07690S. The funders had no role in study design, data collection and analysis, decision to publish, or preparation of the manuscript.

### Grant Disclosures

The following grant information was disclosed by the authors:
Czech Science Foundation under Grant GA 19-07690S.

### Competing Interests

The authors declare that they have no competing interests.

### Author Contributions

- Filip Děchtěrenko conceived and designed the experiments, performed the experiments, analyzed the data, prepared figures and/or tables, authored or reviewed drafts of the paper, and approved the final draft.
- Daniela Jakubková conceived and designed the experiments, performed the experiments, authored or reviewed drafts of the paper, and approved the final draft.
- Jiří Lukavský conceived and designed the experiments, analyzed the data, authored or reviewed drafts of the paper, and approved the final draft.
- Christina J. Howard conceived and designed the experiments, authored or reviewed drafts of the paper, and approved the final draft.

### Human Ethics

The following information was supplied relating to ethical approvals (*i.e.*, approving body and any reference numbers):

All experiments were approved by Ethics committee of Institute of Psychology, Czech Academy of Sciences (approval number PSU-965/Brno/2021).

### Data Availability

Data and scripts are available at Open Science Framework (OSF): Dechterenko, Filip, Daniela Jakubkova, Jiri Lukavsky, and Christina Howard. 2022. "Tracking Multiple Fish." OSF. January 24. DOI 10.17605/OSF.IO/TFRZJ.

Stimuli for the study were manually created using editing filmed videos in Sea world with permission of the management. The rest of the stimuli were created using code in R.

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
