# Peer review of "Tracking multiple fish"

_PeerJ, doi:10.7717/peerj.13031_

## Round 0.1 · original submission · Minor Revisions

Thank you for submitting your manuscript to PeerJ. I have received two reviews, from Ian Thorton and Anna Hughes. I've also read the manuscript myself. You'll be pleased to read that both reviewers are positive about your manuscript, and mainly suggest some minor changes. I agree with the reviewers' assessment: this is an incremental but solid study that highlights the ecological validity of the MOT paradigm. I invite you to submit a revision of the manuscript. Given the minor nature of the comments, I don't intend to send the manuscript out for review again, but of course, I reserve the right to do so if for whatever reason this seems necessary.

·

Basic reporting

This paper presents two experiments that nicely extend previous studies of multiple object tracking (MOT). The question is how a real-world tracking example (fish within a fish tank) relates to standard lab tasks. The research questions are well-motivated and placed appropriately in context. All aspects of the paper conform to a high scientific standard and material is clearly presented.

I seem to recall some previous studies that had specifically examined occlusion in the context of MOT? As this is relevant – and mentioned as a factor here – perhaps chase these down?

Experimental design

Both experiments are clearly described in terms of methods and conform to appropriate ethical standards. Research questions are clear and statistical methods transparent. All material available online.

I like the use of d-prime as the summary of sensitivity. In this context, I wasn’t quite sure what to make of the bias measure. Perhaps help the reader by explaining how bias should be interpreted in the context of MOT? If I can’t tell items apart, how do I select in a biased manner? This becomes relevant in the results as the pattern of higher bias with small tracking size, could just be an artefact, no?

Validity of the findings

This set of findings is very interesting. Extends previous work suggesting that MOT is an ecologically relevant task. Also highlights some novel aspects of real-world tracking that will be useful going forward.

Would it be worth explicitly comparing levels of performance in the three conditions of Exp2? Figure 4 suggests, for example, that standard MOT leads to consistently higher performance than the fakefish condition. Forgive me if I missed the direct comparison, but if you didn’t explore this, might be a worthwhile addition?

Additional comments

Nice work!!

·

Basic reporting

The manuscript is mostly very well written throughout. I spotted a few small typos:
- L26: I think this should be "similarities"
- L83: "by comparing the performance..." - remove "the"
- L142/143: "sudden movements of targets/distractors" might be clearer phrasing
- L172: I think this should be "commonalities"
- L177: the tenses here are a bit confusing, I would probably re-word to "allowed the fish to blend in or even to be temporarily occluded, making them more difficult to detect"
- L295: "that response bias..." - add "the"
- L312: I think this should be "MOTs" i.e. plural
- L316/317: phrasing seems a little awkward here - perhaps "as they are difficult to identify from each other"
- L320: "when number of tracked fish..." - add "the" i.e. "the number of tracked fish..."

The literature review seems good overall. One possible omission is re. L57 - there is some work from the camouflage literature that might be relevant here e.g. https://royalsocietypublishing.org/doi/full/10.1098/rspb.2013.0064 (although the tasks are slightly different from MOT-type tasks).

The article is well structured, and the OSF directory is very complete and easy to use. I had only one minor comment with reference to the figures - in the caption to Figure 4, it might be worth using the phrasing "thick vertical lines" when referring to the 66% high density credible interval, to contrast with the "thin vertical lines" referred to later in the sentence.

There aren't really specific hypotheses given in the manuscript, but the study is clearly described and self-contained.

Experimental design

The work is clearly original, primary research and the extent to which laboratory tasks can predict more real-world behaviour is an important question. The work seems to have been done rigorously and ethics procedures followed.

The methods are clear and stimuli/code have been provided, which would help with replications/extensions.

One small point is that it would be helpful to include a reference for the Bayesian estimation of signal detection models (i.e. around L139) – I ended up reading more about it to understand how sensitivity/bias were calculated, and other readers might find this useful too.

Another very minor comment is that I found the phrasing on L191 "to make the distinction between conditions clearer..." a little odd - as I think you use the 'circle tracking MOT' notation when you want to lump several conditions together, so you are not distinguishing between them! I would suggest removing this phrase.

Validity of the findings

The data and code are provided, and are straightforward to follow (thanks to the authors for providing the R Markdown file, which makes it easy to check which variables in the code correspond to the numbers in the paper!)

My only comment about the conclusions is that in places, they seem a little strong, based on the data - for example:
L348/349: "Although using artificial fish trajectories increased the intra-individual correspondence between tracking accuracies using real and artificial stimuli"
L356-358: "Using motion trajectories in MOT that are derived from a real-world context of fish movement led to better prediction of performance for tracking real fish"
If I have understood the results correctly, there isn't much difference between the standard MOT trajectories and the "fake fish" ones? If this is the case, I would suggest rewording these sections of the discussion slightly to make this clearer.

---

## Round 0.2 · accepted · Accept

Thank you again for your submission to PeerJ. As I already indicated in the previous decision letter, and after having assessed the changes myself, I didn't feel it was necessary to go through another round of reviews. In other words: it is my pleasure to accept your manuscript in its current form for publication in PeerJ. Congratulations!